

# A bi-criterion sequence-dependent scheduling problem with order deliveries

Jian-You Xu[1], Win-Chin Lin[2], Kai-Xiang Hu[2], Yu-Wei Chang[2], Wen-Hsiang Wu[3], Peng-Hsiang Hsu[4], Tsung-Hsien Wu[5] and Chin-Chia Wu[2]

[1] College of Information Science and Engineering, Northeastern University, Shenyang, China
[2] Department of Statistics, Feng Chia University, Taichung, Taiwan
[3] Department of Healthcare Management, Yuanpei University of Medical Technology, Hsinchu, Taiwan
[4] Department of Business Admistration, University of Kang-Ning, Taipei, Taiwan
[5] Bachelor's Program in Business Management, Fu Jen Catholic University, New Taipei City, Taiwan

Corresponding author
Chin-Chia Wu, cchwu@fcu.edu.tw

## ABSTRACT

The manufacturing sector faces unprecedented challenges, including intense competition, a surge in product varieties, heightened customization demands, and shorter product life cycles. These challenges underscore the critical need to optimize manufacturing systems. Among the most enduring and complex challenges within this domain is production scheduling. In practical scenarios, setup time is whenever a machine transitions from processing one product to another. Job scheduling with setup times or associated costs has garnered significant attention in both manufacturing and service environments, prompting extensive research efforts. While previous studies on customer order scheduling primarily focused on orders or jobs to be processed across multiple machines, they often overlooked the crucial factor of setup time. This study addresses a sequence-dependent bi-criterion scheduling problem, incorporating order delivery considerations. The primary objective is to minimize the linear combination of the makespan and the sum of weighted completion times of each order. To tackle this intricate challenge, we propose pertinent dominance rules and a lower bound, which are integral components of a branch-and-bound methodology employed to obtain an exact solution. Additionally, we introduce a heuristic approach tailored to the problem's unique characteristics, along with three refined variants designed to yield high-quality approximate solutions. Subsequently, these three refined approaches serve as seeds to generate three distinct populations or chromosomes, each independently employed in a genetic algorithm to yield a robust approximate solution. Ultimately, we meticulously assess the efficacy of each proposed algorithm through comprehensive simulation trials.

# INTRODUCTION

In today's manufacturing landscape, challenges such as intense competition, a proliferation of product varieties, heightened customization demands, and shortened product life cycles have become more pressing than ever. Consequently, there is greater focus on the need to optimize manufacturing systems. Among the well-established scheduling problems, the customer order scheduling problem (COSP) has always held significant importance in

the realm of manufacturing. The primary challenge of the COSP lies in determining the sequence of jobs to meet the diverse demands of customers ordering various products to be processed on a single machine.

The COSP holds particular significance in make-to-order and make-to-assembly production systems, wherein a single machine is responsible for creating various types of products. In this context, a customer may place an order for one or multiple such products, with the shipment dispatched only after all specified items have been manufactured and packaged together. The concept of COSP was initially introduced by *Julien & Magazine (1990)*. Their work introduced a dynamic programming model and delved into the fundamental characteristics of the optimal solution for the broader case. Over the past decades, COSP on both single and multiple machines has attracted numerous researchers. For instance, *Erel & Ghosh (2007)* addressed COSP with the objective of minimizing the total order lead time and established for the first time that the problem is NP-hard. *Su, Chen & Chen (2013)* considered COSP on parallel machines dispatched in batches to minimize maximum lateness and developed three heuristics based on scheduling rules. *Framinan & Perez-Gonzalez (2018)* proposed a mixed-integer linear programming formulation, a constructive heuristic, and two matheuristics to minimize the total tardiness of orders. *Wu et al. (2019)* addressed COSP with learning effects on multiple machines to minimize total tardiness, subsequently developing four heuristics, three metaheuristics, and a branch-and-bound to solve it.

In practical scenarios, a setup time is required whenever a machine transitions from processing one product to another. Optimizing setup times is an important issue for several reasons, particularly in manufacturing and production environments. For example, shorter setup times mean less downtime between production runs. Reduced setup times enhance a company's ability to respond quickly to changes in customer demand or shifts in the market. Longer setup times often involve increased labor costs, as more time and manpower are required for the setup process. Optimizing setup times is a strategic initiative that can positively impact various aspects of a business, including productivity, flexibility, costs, and overall competitiveness. It aligns with modern manufacturing principles such as lean manufacturing and just-in-time production, which aim to eliminate waste and enhance efficiency throughout the production process. *Allahverdi & Soroush (2008)* emphasized that streamlining setup times or reducing costs contributes to the timely delivery of dependable products or services. For a comprehensive review of setup time applications, readers are directed to the in-depth survey papers authored by *Allahverdi (2015)*, *Yang & Liao (1999)*, *Allahverdi, Gupta & Aldowaisan (1999)*, *Allahverdi et al. (2008)*, *Cheng, Gupta & Wang (2000)*, and other notable works.

Despite the acknowledged significance of setup time in job scheduling, the literature on the COSP with setup time is limited. *Hazır, Günalay & Erel (2008)* conducted a comparative study involving four different metaheuristics to minimize average customer order flow time. Recently, *de Athayde Prata, Rodrigues & Framinan (2021a)* proposed two mixed-integer linear programming models and a fixed variable list algorithm for solving the COSP with sequence-dependent setup time (SDST). Furthermore, they introduced a discrete

differential evolution algorithm for the same problem in *de Athayde Prata, Rodrigues & Framinan (2021b)*.

To align with the reality of customer order scheduling, this study considers the classification of jobs from different orders into distinct classes. Every order consists of at least one job from each job class, necessitating a setup time whenever the machine transitions between classes. In the domain of single-machine scenarios with multiple objectives and various job classes, *Liao (1993)* introduced a branch and bound (B&B) algorithm to systematically investigate all viable solutions. *Gupta, Ho & van der Veen (1997)* were pioneers in investigating COSP with multiple classes and setup times, introducing a bi-criteria problem. They aimed to minimize both makespan and the total carrying costs of customer orders, with carrying costs computed based on the time difference between the completion times of the first and last job in each customer's order. *Lin, Yin & Liu (2013)* addressed a sequence-dependent scheduling problem that incorporates order delivery, presenting a binary integer program and a dynamic programming algorithm. They also introduced tabu search, iterated local search, and genetic algorithm approaches for obtaining approximate solutions. In related studies on customer order scheduling with setup times, *Erel & Ghosh (2007)* examined customer orders with varying quantities of products from different product families processed on a continuously available machine in any sequence. *Liu (2009)*, *Liu (2010)* and *Hsu & Liu (2009)* explored various scenarios involving multiple jobs within a job shop setting. Recently, *Li et al. (2023)* expanded upon the work laid down by *Gupta, Ho & van der Veen (1997)* by introducing an innovative approach that incorporates four heuristics, three locally enhanced search methods, and a branch and bound algorithm. Additionally, they applied a water wave optimality algorithm, offering four distinct wavelength variants. This collaborative effort is aimed at minimizing a mixed criterion including the order ranges, order tardiness, and total job completion times. In a parallel effort, *Lin et al. (2023)* presented four heuristics complemented by a local search method, four theoretical simulated annealing techniques, a cloudy theoretical simulated annealing hyperheuristic, and a B&B method to solve the problem. Their collective endeavors focused on minimizing a mixed criterion including the total order ranges and total order tardiness. Meanwhile, *Gupta et al. (2023)* introduced six two-phase heuristics, six variants of water-wave optimization algorithms, a mixed-integer linear programming formulation, and a B&B algorithm to solve their problem. This concerted effort is directed at producing approximate solutions that effectively minimize a mixed criterion of the cumulative holding costs for all orders and makespan for all jobs.

Setup times present a significant concern for manufacturing companies. Lengthy setup times can jeopardize the on-time delivery of customer orders, leading to potential losses in customer satisfaction and incurring direct or indirect costs. Effectively incorporating and managing the setup time factor in scheduling decisions can yield substantial benefits (*Kim & Bobrowski, 1997*; *Allahverdi, Gupta & Aldowaisan, 1999*; *Allahverdi et al., 2008*; *van Donk & van Doorne, 2016*; *Zhao et al., 2018*; *Ying et al., 2023*).

Motivated by the scarcity of research that address sequence-dependent bi-criteria scheduling problems that incorporate order deliveries, we introduce a novel problem in this domain. Our aim is to pinpoint a schedule that optimally balances the makespan and the

weighted completion time of designated orders through a linear combination. Expanding on these insights, this study addresses a sequence-dependent bi-criteria scheduling challenge involving order deliveries. The primary aim is to devise an optimized schedule that minimizes a mixed criterion of the overall job makespan and the aggregated weighted completion time of all orders.

The main contributions of this study are summarized as follows: (a) we address a sequence-dependent bi-criteria scheduling challenge involving order deliveries; (b) we propose three lemmas and a lower bound in a branch-and-bound algorithm for finding an optimal solution; (c) we propose a simple heuristic according to the smallest first value of $d_{uv}$ and their three improvements by three local searching methods and (d) we build three population-based genetic algorithms. The subsequent sections of this study are structured as follows. 'Problem Statement' outlines the proposed problem formulation. In 'Heuristic methods and heuristic-based genetic algorithm' , we present three dominant properties, a proposed lower bound, a heuristic, and its three improved schemes (*i.e.,* pairwise interchange, extraction and forward-shifted reinsertion, and extraction and backward-shifted reinsertion). Additionally, we propose three variants of the genetic algorithm (GA) based on these three improved heuristics. 'Tuning the Related Parameter of the Three Heuristic-Based Genetic Algorithms' focuses on testing the appropriate parameter values in the GA. 'Computational Simulations and Discussions' provides a comprehensive report of all test results. The final section offers conclusions and outlines future research directions.

## PROBLEM STATEMENT

The formal framework of our proposed study is outlined as follows. Let us consider a set $\Omega = O_1, \ldots, O_m$ comprising $m$ orders designated for processing on a single machine. The machine cannot break down while performing all jobs. Each order $O_i$ consists of $n_i$ jobs, each assigned a weight $w_i$ ($w_i \in (0, 1)$). The total number of jobs across all orders amounts to $n$, which defines the set $N = \{J_1, \ldots, J_n\}$.

A critical assumption In this model (see examples in *Lin, Yin & Liu (2013)*) is that, irrespective of their respective orders, if job $J_v$ immediately follows job $J_u$ in the schedule, then there is a sequence-dependent setup time $d_{uv} > 0$ required. Additionally, we define $d_{0J_u}$ to signify the machine status before the initial scheduling of job $J_u$, elucidating the setup required.

Notably, in this model, we assume that the processing times for all jobs in set $N$ are negligible. Given a complete schedule $\sigma$ encompassing all jobs, we introduce the notation $C_{Op}(\sigma)$ to represent the completion time for order $O_p$, which is the time at which the last job of $O_p$ is finished. Simultaneously, the makespan, denoted as $C_{max}(\sigma)$, signifies the completion time of the last job in set $N$.

The primary objective of this study is to determine an optimal schedule that minimizes a linear combination of the sum of makespan and the weighted completion times of all $m$ orders. In mathematical terms, we aim to find a solution to the following optimization problem:

Minimize $g(\sigma) = \theta C_{max}(\sigma) + (1 - \theta)\sum_{p=1}^{m} w_p C_{Op}(\sigma).$

Here, $\theta$ represents a weighting parameter that allows for the adjustment of the trade-off between makespan and weighted completion times.

To simplify matters, we use the terms "node," "job," and "city" interchangeably to describe the fundamental elements of our scheduling problem. Additionally, we introduce two notations, $C_u(\sigma)$ and $C_v(\sigma)$ to represent the completion times of jobs $u$ and $v$ in schedule $\sigma$, respectively. $C_v(\sigma')$ and $C_u(\sigma')$ represent the completion times of jobs $v$ and $u$ in schedule $\sigma'$, respectively, where $\sigma = (\pi, J_u, J_v, \pi^c)$ and $\sigma' = (\pi, J_v, J_u, \pi^c)$. Here, $\pi$ and $\pi^c$ are two distinct sequences within set $N$.

To enhance the efficiency of our B&B method, we establish two properties that assist in narrowing down the nodes that satisfy the conditions outlined in these properties. Before presenting the properties, we designate job $J_L$ as the final job in sequence $\pi$.

**Property 1**: As jobs $J_u \in O_p$, $J_v \in O_q$, $p \neq q$, if $J_u$ is the last assigned job of $O_p$, and $J_v$ is the last assigned job of $O_q$, $d_{Lu} + \frac{w_q}{w_p + w_q} d_{uv} < d_{Lv} + \frac{w_p}{w_p + w_q} d_{vu}$ and $d_{Lu} + d_{uv} + max_{x \in \pi^c}\{d_{vx}\} < d_{Lv} + d_{vu} + min_{x \in \pi^c}\{d_{ux}\}$, then $\sigma$ dominates $\sigma'$.

Proof: In following, one will claim that (i) $C_{max}(\sigma') - C_{max}(\sigma) > 0$, and (ii) $\sum_{p=1}^{m} w_p C_{Op}(\sigma') - \sum_{p=1}^{m} w_p C_{Op}(\sigma) > 0$. Then, $g(\sigma') - g(\sigma) > 0$.

Let $t_\pi$ be the completion time of the last job in $\pi$, and $t_{p1}$ ($t_{q1}$) be the completion time of the first job of $O_p$ ($O_q$). Note that $t_{p1} < t_\pi$ ($t_{q1} < t_\pi$). Moreover, for any possibly arranged job sequence of all jobs in $\pi^c$, let $J_s$ be the first assigned job in $\pi^c$. Note that $J_s \neq J_u$, $J_s \neq J_v$. Then, one has the following results, (i) and (ii).

(i) $C_{max}(\sigma') - C_{max}(\sigma) = [d_{Lv} + d_{vu} + d_{us}] - [d_{Lu} + d_{uv} + d_{vs}] > 0$. The last inequality follows from the given condition $d_{Lu} + d_{uv} + max_{x \in \pi^c}\{d_{vx}\} < d_{Lv} + d_{vu} + min_{x \in \pi^c}\{d_{ux}\}$.

(ii) For the orders $O_p$ and $O_q$, $[w_q C_{Oq}(\sigma') + w_p C_{Op}(\sigma')] - [w_p C_{Op}(\sigma) + w_q C_{Oq}(\sigma)] = \left[ w_q(t_\pi + d_{Lv} - t_{q1}) + w_p(t_\pi + d_{Lv} + d_{vu} - t_{p1}) \right] - \left[ w_p((t_\pi + d_{Lu} - t_{p1})) + w_q(t_\pi + d_{Lu} + d_{uv} - t_{q1}) \right] = w_q[d_{Lv} - d_{Lu} - d_{uv}] + w_p[d_{Lv} + d_{vu} - d_{Lu}] = (w_q + w_p)\left[ d_{Lv} - d_{Lu} + \frac{w_p}{w_q + w_p} d_{vu} - \frac{w_q}{w_q + w_p} d_{uv} \right] > 0$.

For these orders (not including $O_p$ and $O_q$), which are the last jobs of the order assigned in $\pi^c$, the sum of all $w_p C_{Op}(\sigma') - w_p C_{Op}(\sigma)$ is $\sum_p w_p C_{Op}(\sigma') - \sum_p w_p C_{Op}(\sigma) = \sum_p w_p[(d_{Lv} + d_{vu} + d_{us}) - (d_{Lu} + d_{uv} + d_{vs})] > 0$. Therefore, $\sum_{p=1}^{m} w_p C_{Op}(\sigma') - \sum_{p=1}^{m} w_p C_{Op}(\sigma) > 0$. Combining the results of (i) and (ii), $g(\sigma') > g(\sigma)$ follows. $\square$

**Property 2**: As jobs $J_u \in O_p$, $J_v \in O_q$, $p \neq q$, if $J_u$ is not the last assigned job of $O_p$, and $J_v$ is also not the last assigned job of $O_q$, $d_{Lu} + d_{uv} < d_{Lv} + d_{vu}$ and $max_{x \in \pi^c} d_{vx} < min_{y \in \pi^c} d_{uy}$, then $\sigma$ dominates $\sigma'$.

Proof: It needs to be shown that $g(\sigma) < g(\sigma')$.

For any possibly arranged job sequence of all jobs in $\pi^c$, one can be shown that $C_{max}(\sigma) < C_{max}(\sigma')$, and $\sum_{p=1}^{m} w_p C_{Op}(\sigma) - \sum_{p=1}^{m} w_p C_{Op}(\sigma') < 0$.

Let $J_s$ be the first assigned job in $\pi^c$. Note that $J_s \neq J_u$, $J_s \neq J_v$. Then, $C_{max}(\sigma) - C_{max}(\sigma') = [d_{Lu} + d_{uv} + d_{vs}] - [d_{Lv} + d_{vu} + d_{us}] = [(d_{Lu} + d_{uv}) - (d_{Lv} + d_{vu})] + (d_{vs} - d_{us})$. Applying two given inequalities, one has $C_{max}(\sigma) - C_{max}(\sigma') < 0$. By the same argument, $\sum_{p=1}^{m} w_p C_{Op}(\sigma) - \sum_{p=1}^{m} w_p C_{Op}(\sigma') = \sum_p w_p[(d_{Lu} + d_{uv}) - (d_{Lv} + d_{vu}) + (d_{vs} - d_{us})] < 0$ because $0 < w_p < 1$. The last sum is over all orders on which the last job of the order is in $\pi^c$. Thus, $g(\sigma) < g(\sigma')$. $\square$

**Property 3**: Consider a scheduled $\sigma = (PS, US)$, where $PS$ and $US$ denote the scheduled part with $k$ jobs and the unscheduled part with $(n$-$k)$ jobs, and $PS \cup US = N$. If

$d_{J_{[k-1]}J_{[k]}} > d_{J_{[k-1]}J_v}$ for all $J_v \in US$, and $J_v$ and $J_{[k]}$ belong to the same order, then $\sigma = (PS, US)$ can be eliminated.

In the ensuing discussion, our objective is to establish a lower bound for the schedule denoted as $\sigma = (\pi, \pi^c)$, where $\pi$ represents the scheduled portion comprising $k$ jobs, and $\pi^c$ signifies the set of $n$-$k$ unscheduled jobs. Let $t_\pi$ denote the completion time of the final job, $J_L$, in $\pi$, and $\sum_{i \in \pi} w_i C_{O_i}$ stand for the aggregate weighted completion time of the scheduled orders in $\pi$.

In light of the sequence-dependent aspect, we incorporate $J_L L$ into the set $\pi^c$, resulting in the creation of a new set denoted as $\pi^{c*} = \pi^c \cup \{J_L\}$, now comprising $(n-k+1)$ jobs. To approximate the completion times in $\pi^c$, we embark on the following procedure to extract the $(n-k)$ smallest sequence-dependent setup times within $\pi^{c*}$. These recorded values are denoted as D(1), D(2), ..., D($n-k$).

```
01:  do u = 1, (n − k + 1)
02:  do v = 1, (n − k + 1)
03:      if (u ≠ v) then
04:          index(Jᵤ, Jᵥ) = 1 for Jᵤ, Jᵥ ∈ π^c*
05:      endif
06:  enddo
07:  enddo
08:  l = 0
09:  do i = 1, (n − k)
10:      idmin = M (a very large number)
11:      do u = 1, (n − k + 1)
12:      do v = 1, (n − k + 1 )
13:          if (d_{JᵤJᵥ} < idmin and index(Jᵤ, Jᵥ) = 1) then
14:              idmin = d_{JᵤJᵥ}
15:              id1 = Jᵤ
16:              id2 = Jᵥ
17:          endif
18:      enddo
19:      enddo
20:  l = l + 1
21:  set D(l) = idmin; index(id1, id2) = 0
22:  enddo
23:  output D(1), ..., D(n − k).
```

Notably, the values of D(1), ..., D($n-k$) exhibit a non-decreasing trend. Consequently, we can estimate the lower bound for each $C_{[k+i]}(\sigma)$, with $i$ ranging from 1 to $n-k$, by employing the following formula:

$$C_{[k+i]}(\sigma) = t_\pi + \sum_{v=1}^{i} D_v, i = 1, \ldots, n-k, \tag{1}$$

Additionally, we maintain a record of the frequency number $f_i$ corresponding to each order in $\pi^c$, where $i$ spans from 1 to $q$, with $1 \leq f_1 \leq f_2 \leq \ldots \leq f_q$ and $q < m$. Utilizing Eq. (1),

we derive completion times for the outstanding orders in $\pi^c$ as $C_{[k+f_1]}(\sigma)$, $C_{[k+f_1+f_2]}(\sigma)$, ..., $C_{[k+f_1+f_2+...+f_q]}(\sigma)$.

Furthermore, we arrange the weights $w_{(1)} \le w_{(2)} \le ... \le w_{(q)}$ in non-decreasing order within the set $\{w_v \in \pi^c, v = 1, ..., q\}$. Applying the lemma established by *Hardy, Littlewood & Polya (1967)*, we arrive at a lower bound for the estimated weighted completion times of the unscheduled $q$ orders, represented as $\sum_{i=1}^{q} w_{(q-i+1)} C_{\left[k+\sum_{v=1}^{i} f_v\right]}(\sigma)$. Hence, the lower bound is derived as follows:

$$\theta C_{max}(\sigma) + (1-\theta)\sum_{p=1}^{m} w_p C_p(\sigma) \ge \theta\left(t_\pi + \sum_{v=1}^{n-k} D_v\right) +$$

$$(1-\theta)\left(\sum_{v=1}^{b} w_v C_{[v]}(\sigma) + \sum_{i=1}^{q} w_{(q-i+1)} C_{\left[k+\sum_{v=1}^{i} f_v\right]}(\sigma)\right), \text{ where } b+q=m.$$

## Heuristic methods and heuristic-based genetic algorithm

Given that the proposed problem with only a single criterion has been shown to be NP-hard, the problem require significant CPU time to find exact solutions as the number of jobs increases. Thus, three heuristics based on problem characteristics will be provided. In the beginning, we choose a job $J_u$ with the smallest value among job set $N$, that is, $d_{0J_u} = \min\{d_{0J_i}, J_i \in N\}$ to be scheduled on the first position. Subsequently, we choose the second job $J_v$ with the smallest value among job set $N \setminus \{J_u\}$; *i.e.*, $d_{J_u J_v} = \min\{d_{J_u J_i}, J_i \in N \setminus \{J_u\}\}$, and we perform the same process until all jobs are determined to yield an initial schedule $\sigma$. To obtain a finer quality of solutions, following the idea of *Della Croce, Narayan & Tadei (1996)*, we perform three improved schemes (pairwise interchange (PI), extraction and forward-shifted reinsertion (FOR) and extraction and backward-shifted reinsertion (BK)) on the initial schedule $\sigma$. They are denoted as DPI, DFOR, and DBK. The details of the proposed heuristics are given as follows:

*The procedures of three improved algorithms*:

However, the existing level of complexity is insufficient to address the intricacies of the proposed model. Conversely, metaheuristics present a notably higher level of complexity and present a significant challenge in terms of effective design and implementation, especially in the context of intelligent random search strategies (*Holland, 1975*; *Bean, 1994*; *Larranaga et al., 1999*; *Nguyen, Mei & Zhang, 2017*).

Within the realm of metaheuristics, the GA distinguishes itself as a well-established and effective method for generating high-quality approximate solutions to a diverse set of combinatorial problems. Extensive research has demonstrated the GA's ability to discover near-optimal solutions for a wide array of complex challenges (*Iyer & Saxena, 2004*; *Essafi, Matib & Dauzere-Peres, 2008*; *Beasley, Bull & Martin, 1993*; *Fan et al., 2022*).

The foundational principles of GA were rigorously established by *Holland (1975)*. A genetic algorithm initiates with a set of feasible solutions, referred to as the population, and subsequently replaces the current population through an iterative process. This entails the necessity of a suitable encoding for the specific problem and the formulation of a fitness function that serves as a metric for assessing the quality of each encoded solution (chromosome or individual). The reproductive mechanism selects the parents and employs a crossover operator to recombine them, resulting in offspring that undergo a mutation

00: Input $\{d_{0J_i}, J_i \in N\}$ and $\{d_{J_iJ_j}, J_i, J_j \in N\}$

01: Set $\sigma$ as an empty set and $l = 1$

02: Choose $J_u$ with the smallest value of $d_{0J_u}$ among $\{d_{0J_i}, J_i \in N\}$ to be the $l^{\text{th}}$ position

03: Set $\sigma = (J_u, \ldots)$, delete $J_u$ from $N$, record as $N \backslash \{J_u\}$, and $l = l + 1$

04: Do while $\{l <= n\}$

05:     Choose $J_v$ with the smallest value of $d_{J_uJ_v}$ among $\{d_{J_uJ_i}, J_i \in N \backslash \{J_u\}\}$ to be the $l^{\text{th}}$ position

06:     Set $\sigma = (J_u, J_v, \ldots)$, delete $J_v$ from $N\{J_u\}$

07:     Record as $N \backslash \{J_u, J_v\}$, and $l = l + 1$

08: End while.

09: Output the final solution $\sigma$.

10: Apply the methods (PI, FOR, and BK) separately to improve $\sigma$

11: Do $k_1 = 1$ to $n - 1$

12:     Do $k_2 = k_1 + 1$ to $n$

13:         Choose the $k_1$ position job and the $k_2$ position job in $\sigma$ to perform the PI, FOR, and BK separately to improve $\sigma$.

14:         Keep the best one.

15:     End do

16: End do

17: Output these three refined solutions (DPI, DFOR, and DBK).

operator to introduce local alterations (*Essafi, Matib & Dauzere-Peres, 2008*). The genetic algorithm proceeds through the following key stages:

**Step 1. Representation of structure**:

▶A structure is defined as a sequence of jobs (*Etiler et al., 2004*)

**Step 2. Initial population**:

▶Drawing on established heuristics in the literature is advised for constructing the initial population (*Etiler et al., 2004*). This approach expedites the convergence toward the final solution. This study adopts three initial sequences:

▶DPIGA, DFORGA, and DBKGA create their populations based on three initial sequences DPI, DFOR, and DBK by using the pairwise interchange method.

**Step 3. Population size**:

▶The population is set at nsize.

**Step 4. Fitness function**:

▶The objective is to minimize the *objective function $g(\sigma)$* (*i.e.*, a linear combination of the sum of the makespan and the weighted completion times of all *m* orders, *i.e.*, $g(\sigma) = \theta C_{max}(\sigma) + (1 - \theta)\sum_{i=1}^{m} w_i C_i(\sigma)$. The fitness function of each string (*i.e.*, *i*th string in the *t*th generation) is calculated as

$h(\sigma_i(t)) = max_{1 \le j \le nsize} \{g(\sigma_j(t))\} - g(\sigma_i(t))$

and the probability of selection of the *i*th string is defined as

$P(\sigma_i(t)) = h(\sigma_i(t)) / \sum_{i=1}^{nsize} h(\sigma_i(t))$

▶This function is pivotal for parent selection and reproduction.

**Step 5. Crossover**:

▶This study employs the partially matched crossover technique, with a crossover rate of 1

**Step 6. Mutation**:

▶The mutation rate is set at $p$.

**Step 7. Selection**:

▶Population sizes remain constant at nsize from one generation to the next. Offspring, excluding the best schedule with the minimum $g(\sigma)$, are generated from parent chromosomes using the roulette wheel method.

**Step 8. Stopping rule**:

▶Each DPIGA, DFORGA, or DBKGA iteration is concluded after gsize times.

## TUNING THE RELATED PARAMETER OF THREE HEURISTIC-BASED GENETIC ALGORITHMS

In this section, we conduct several experimental tests to explore the values of related parameters of three heuristic-based genetic algorithms (GAs). To determine the parameters used in three GAs as chromosome count *nsize*, evolution generations *gsize* and mutation rate *p*, we consider an experimental design as the number of jobs at $n = 12$, the number of orders at $m = 2$, and the weight of objective function at $\theta = 0.25$. One hundred instances are tested for this case. The initial setup times $d_{0J_u}$ and sequence-dependent setup time $d_{uv}$ are generated from uniformly distributed U[1, 20], while the weights of orders $w_i$ are generated from uniformly distributed U[1, 10]. The optimal solutions ($Opt_i$) and the near-optimal solutions ($GAH_i$) are produced by performing the B&B method and three heuristic-based GAs (DPIGA, DFORGA, or DBKGA). The measurement criterion is based on the average error percentage (AEP) or $AEP = \left\{ \sum_{i=1}^{100} \left( GAH_i - Opt_i \right) / Opt_i \right\} / 100$. The parameter tuning process is based on a single-factor experimental method applied to a set of generated problem instances.

To evaluate the value of *nsize*, *gsize* was fixed initially at 50 and *p* was fixed at 0.02. We executed DPIGA, DFORGA, and DBKGA several times and changed the values of *nsize* over the interval [10, 50] with an increment of four. As shown in Fig. 1 (top), as the *nsize* value increases, the trends of the AEPs rapidly decrease and become flat at point 50 obtained from DBKGA. Therefore, in subsequent simulation experiments, the *nsize* is adopted at 50.

With the previously determined *nsize* at 50 and fixed mutation rate at 0.02, we performed three GAs to determine the values of *gsize* over [10, 200] with an increment of 10. Figure 1 (middle) shows that the AEP of DBKGA has a minimum point at 180. Thus, *gsize* is set at 180 for all three proposed GAs in later experiments.

After investigating the parameters *nsize* and *gsize*, we established *nsize* = 50 and *gsize* = 180. We conducted three GAs to assess the mutation rate *p*, ranging from 0.01 to 0.2 with a consistent increment of 0.01. As depicted in Fig. 1 (bottom), AEP demonstrates a gradual convergence, while DBKGA attains the minimal error at a mutation rate of 0.2. Accordingly, we adopt a mutation rate of 0.2 as the benchmark for all three GAs in the subsequent stages.

In a similar way, we conducted another one hundred random instances at $n = 240$ to explore the values of three parameters of *nsize*, *gsize*, and *p* for three GAs (DPIGA, DFORGA, and DBKGA). Based on previous studies, to capture parameters that outperform three

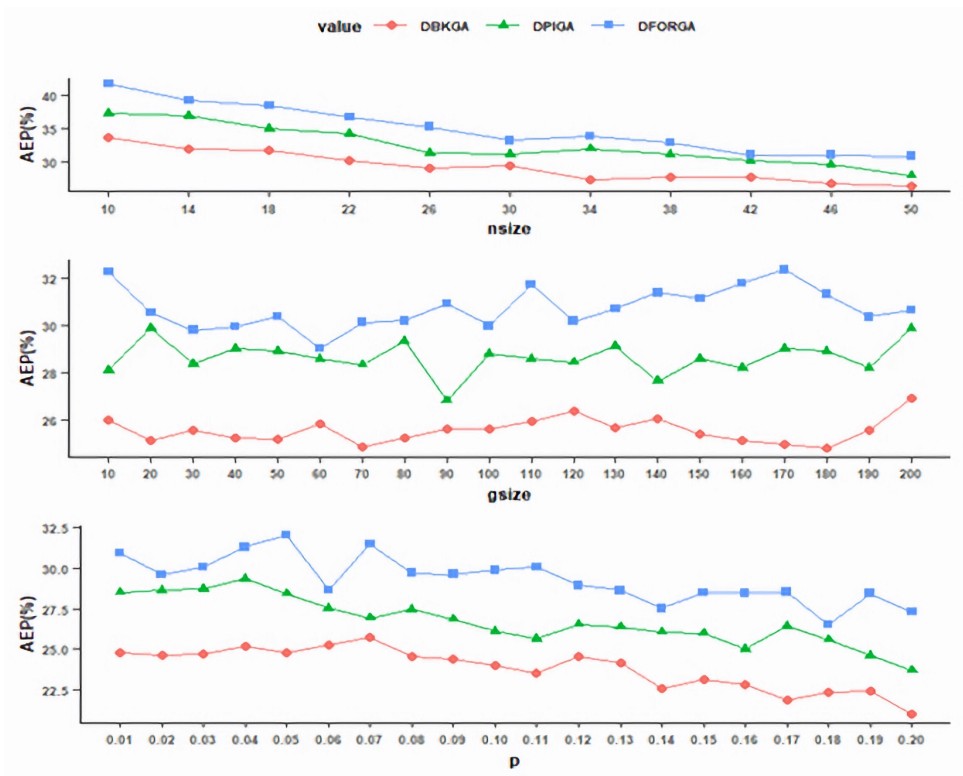

**Figure 1  Exploring the values of parameters for three GAs.**

simple heuristic algorithms (DPI, DFOR, and DBK), we only performed DFORGA to change the values of one parameter but fixed other parameters each time and compared the objective function of DFORGA with those from DPI, DFOR, and DBK. In this case, we performed three simple heuristic algorithms (DPI, DFOR, and DBK) and DFORGA to solve the data set and computed each average of their objective functions. The final results are summarized and reported in Table 1. It can be observed in Table 1 that $nsize = 100$, $gsize = 1500$, and $p = 0.2$ are appropriate values for using DPIGA, DFORGA, and DBKGA.

Based on the above tests, the population size ($nsize$), generation size ($gsize$), and mutation rate ($p$) are set at (50, 180, 0.2) for the small n case and at (100, 1500, 0.2) for big n case in all three GAs (DPIGA, DFORGA, and DBKGA).

## COMPUTATIONAL SIMULATIONS AND DISCUSSIONS

This section conducts some tested designs to assess the performances of the proposed B&B, three heuristic methods (DPI, DFOR, and DBK), and three heuristic-based GAs (DPIGA, DFORGA, and DBKGA). The solution methodologies for the proposed problem are summarized as follows.

Xu et al. (2024), *PeerJ Comput. Sci.*, DOI 10.7717/peerj-cs.1763

**Table 1  The summary of changing related parameters in DFORGA.**

| | DPI | DFOR | DBK | DFORGA | DFORGA | DFORGA | DFORGA | DFORGA |
|---|---|---|---|---|---|---|---|---|
| (*nsize, gsize, p*) | | | | (100, 1500, 0.2) | (150, 1500, 0.2) | (200, 1500, 0.2) | (250, 1500, 0.2) | (300, 1500, 0.2) |
| mean | 7076.30 | 7144.24 | 7072.63 | 7069.48 | 7076.26 | 7101.08 | 7125.53 | 7128.32 |
| (*nsize, gsize, p*) | | | | (100, 1200, 0.2) | (100, 1300, 0.2) | (100, 1400, 0.2) | (100, 1500, 0.2) | (100, 1600, 0.2) |
| mean | 7076.30 | 7144.24 | 7072.63 | 7079.52 | 7077.23 | 7072.63 | 7069.48 | 7071.43 |
| (*nsize, gsize, p*) | | | | (100, 1500, 0.16) | (100, 1500, 0.17) | (100, 1500, 0.18) | (100, 1500, 0.19) | (100, 1500, 0.2) |
| mean | 7076.30 | 7144.24 | 7072.63 | 7085.795 | 7072.375 | 7072.69 | 7084.76 | 7069.48 |

| Optimal solution method | B&B |
|---|---|
| Approximate solution methods | DPI, DFOR, and DBK |
| | DPIGA, DFORGA, and DBKGA |

All the related parameters are designed as follows. The initial setup times $d_{0J_u}$ and sequence-dependent setup time $d_{uv}$ are generated from uniformly distributed U[1, 20], and the weights of orders $w_i$ are generated from uniformly distributed U[1, 10]. The experimental designs include the weight of the objective function at $\theta = 0.25$, 0.5, and 0.75, the number of jobs at $n =$12, 16, 20, and 24 for the small-job case, and $n =$60, 120, 180 and 240 for the large-number-of-job case. It should be emphasized that $n =m^\star nc$ signifies that the total of $n$ jobs can be apportioned among $m$ orders, with each order containing $nc$ jobs. The details are summarized in Table 2. The experiment investigates 24 cases for both small and large job numbers. In each case, 100 instances are generated to assess the performance of all proposed methods. In total, 2,400 instance problems are examined for both small and large job sets. The findings are summarized in Tables 3, 4, and 5.

When dealing with a case with a small number of jobs, we assess the effectiveness of the three heuristic algorithms (DPI, DFOR, and DBK) and the DPIGA, DFORGA, and DBKGA algorithms by examining their average error percentage and maximum error across various values of $n$, $m$, and $\theta$. The AEP is computed as follows: $AEP = \left\{ \sum_1^{100} \left( Hsol_i - Opt_i \right) / Opt_i \right\} / 100$, where $Opt_i$ is derived from the B&B method, while $Hsol_i$ isproduced using the six proposed approximate techniques. Additionally, we furnish the mean and maximum values of the explored nodes, along with their respective CPU times (measured in seconds), to evaluate the performance of B&B for each scenario. It is worth noting that an instance problem is considered feasible if the B&B process requires fewer than $10^8$ nodes.

With regard to the performance of B&B, columns 3 and 5 of Table 3 show that B&B consumes more nodes or CPU time at smaller order numbers ($m =$2) compared to larger order numbers ($m =$3 or 4). This implies that the properties and lower bounds become more useful as the number of orders increases. The effect of $\theta$ in columns 3 and 5 of Table 3 does not show a clear trend. Additionally, the number of nodes increases significantly as the number of jobs ($n$) increases. This phenomenon aligns with the characteristics of the NP-hard problem.

In regard to the obtained results for the three heuristics (DPI, DFOR, and DBK) and three genetic algorithms (DPIGA, DFORGA, and DBKGA) applied to cases with small $n$, Table 4 and Fig. 2 present their respective average error percentages. To summarize, the mean AEP values are 0.4733, 0.5246, and 0.4551 for DPI, DFOR, and DBK and 0.3412, 0.3504, and 0.3358 for DPIGA, DFORGA, and DBKGA, respectively. The DBKGA algorithm performed better than all other approaches. Table 4 provides the mean AEP values corresponding to different values of parameters $\theta$, $m$, and $n$. Upon inspection of Table 4, it is evident that the AEPs for the three heuristics and three GAs experience a marginal rise as $n$ increases

**Table 2 The data set of stimulation design.**

| parameter | $n = m * nc$ |
|---|---|
| $n$ | 12=2*6,12= 3*4,16= 2*8,16= 4*420=2*10,20= 4*5, 24=2*12, 24=4*6 |
| | 60=4*15, 60=6*10, 120=6*20, 120=10*12, 180=6*30, 180=10*18, 240=6*40, 240=10*24 |
| $\theta$ | 0.25, 0.5, 0.75 |
| $d_{0Ju}$ | uniformly distributed U[1, 20] |
| $d_{uv}$ | uniformly distributed U[1, 20] |
| $w_i$ | uniformly distributed U[1, 10] |

**Table 3 The results of the B&B method.**

| | | node | | CPU-time | | |
|---|---|---|---|---|---|---|
| $n$ | $m$ | mean | max | mean | max | Total FS |
| 12 | 2 | 3603 | 23080 | 0.09 | 0.39 | 300 |
| | 3 | 2738 | 14983 | 0.07 | 0.30 | 300 |
| 16 | 2 | 94499 | 674705 | 4.57 | 28.60 | 300 |
| | 4 | 39709 | 266953 | 2.27 | 14.09 | 300 |
| 20 | 2 | 1976143 | 13814903 | 216.12 | 1445.10 | 300 |
| | 4 | 754438 | 6457332 | 101.95 | 805.16 | 300 |
| 24 | 2 | 21693624 | 87773444 | 4728.86 | 17225.56 | 176 |
| | 4 | 13871371 | 80046608 | 3447.21 | 18942.06 | 291 |
| | $\theta$ | | | | | |
| 12 | 0.25 | 3250 | 16567 | 0.08 | 0.30 | 200 |
| | 0.50 | 2910 | 16983 | 0.08 | 0.32 | 200 |
| | 0.75 | 3352 | 23545 | 0.08 | 0.41 | 200 |
| 16 | 0.25 | 73172 | 407531 | 3.71 | 20.28 | 200 |
| | 0.50 | 73943 | 666187 | 3.70 | 28.70 | 200 |
| | 0.75 | 54198 | 338770 | 2.84 | 15.06 | 200 |
| 20 | 0.25 | 1588482 | 13093062 | 181.34 | 1368.09 | 200 |
| | 0.50 | 1254223 | 7857226 | 147.42 | 879.21 | 200 |
| | 0.75 | 1253167 | 9458065 | 148.35 | 1128.09 | 200 |
| 24 | 0.25 | 19043167 | 93142508 | 4370.00 | 20408.00 | 183 |
| | 0.50 | 14440245 | 88944864 | 3553.43 | 19576.63 | 98 |
| | 0.75 | 15884964 | 69785019 | 3696.21 | 15540.97 | 186 |
| Total mean | | 3844723 | | 852.64 | | |

from 12 to 24. In contrast, these AEPs exhibit a comparatively stable trend when subjected to changes in the values of $m$ or $\theta$.

According to the criterion of average relative percentage deviation (RPD), RPD is calculated as follows: $RPD = \left\{ \sum_1^{100} (Hsol_i - H_{min})/H_{min} \right\}/100$, where $Hsol_i$ represents the output obtained using the six proposed approximate methods, and $H_{min}$ is the minimum value among these six solutions. Table 5 and Fig. 3 present an evaluation of the performance of the heuristics and genetic algorithms in the context of larger $n$ values. The data in Table

**Table 4  The results of three heuristics (DPI, DFOR, and DBK) and three GAs (DPIGA, DFORGA, and DBKGA).**

| | | DPI | | DFOR | | DBK | | DPIGA | | DFORGA | | DBKGA | |
|---|---|---|---|---|---|---|---|---|---|---|---|---|---|
| $n$ | $m$ | mean | max | mean | max | mean | max | mean | max | mean | max | mean | max |
| 12 | 2 | 0.3628 | 1.1018 | 0.4169 | 1.4138 | 0.3477 | 1.3154 | 0.2307 | 0.7681 | 0.2348 | 0.8269 | 0.2289 | 0.7269 |
| | 3 | 0.3565 | 0.9779 | 0.4097 | 1.4292 | 0.3409 | 1.1221 | 0.2197 | 0.7931 | 0.2243 | 0.7955 | 0.2140 | 0.7603 |
| 16 | 2 | 0.4482 | 1.1769 | 0.4980 | 1.4854 | 0.4433 | 1.2741 | 0.3273 | 0.9767 | 0.3397 | 0.8941 | 0.3282 | 0.8980 |
| | 4 | 0.4388 | 1.3809 | 0.4839 | 1.6221 | 0.4247 | 1.3785 | 0.3173 | 1.0354 | 0.3246 | 1.0215 | 0.3138 | 0.9138 |
| 20 | 2 | 0.4978 | 1.2862 | 0.5535 | 1.4642 | 0.4699 | 1.2807 | 0.3596 | 1.0299 | 0.3705 | 0.9695 | 0.3491 | 0.8306 |
| | 4 | 0.5263 | 1.8886 | 0.5890 | 1.6876 | 0.5108 | 1.4705 | 0.3801 | 1.0245 | 0.3870 | 1.1743 | 0.3761 | 1.0183 |
| 24 | 2 | 0.5799 | 1.2430 | 0.6258 | 1.3743 | 0.5519 | 1.2469 | 0.4418 | 1.0662 | 0.4550 | 0.9625 | 0.4303 | 0.9644 |
| | 4 | 0.5771 | 1.1681 | 0.6211 | 1.6622 | 0.5524 | 1.1642 | 0.4538 | 0.9000 | 0.4681 | 0.9188 | 0.4466 | 0.8626 |
| | $\theta$ | | | | | | | | | | | | |
| 12 | 0.25 | 0.3461 | 0.9620 | 0.3876 | 1.1592 | 0.3139 | 1.0635 | 0.2178 | 0.7679 | 0.2085 | 0.7535 | 0.2035 | 0.6884 |
| | 0.5 | 0.3699 | 1.2663 | 0.4293 | 1.8764 | 0.3639 | 1.6117 | 0.2247 | 0.8055 | 0.2313 | 0.9864 | 0.2290 | 0.7887 |
| | 0.75 | 0.3630 | 0.8914 | 0.4231 | 1.2290 | 0.3551 | 0.9812 | 0.2332 | 0.7684 | 0.2489 | 0.6938 | 0.2319 | 0.7538 |
| 16 | 0.25 | 0.4348 | 1.0311 | 0.4722 | 1.3818 | 0.4100 | 1.2300 | 0.3101 | 0.8895 | 0.3094 | 0.8825 | 0.2993 | 0.7854 |
| | 0.5 | 0.4365 | 1.2126 | 0.4982 | 1.4653 | 0.4325 | 1.1328 | 0.3092 | 0.8963 | 0.3299 | 0.9468 | 0.3167 | 0.8663 |
| | 0.75 | 0.4593 | 1.5929 | 0.5025 | 1.8142 | 0.4596 | 1.6161 | 0.3477 | 1.2324 | 0.3571 | 1.0443 | 0.3471 | 1.0660 |
| 20 | 0.25 | 0.4957 | 1.9983 | 0.5483 | 1.5351 | 0.4710 | 1.2238 | 0.3512 | 0.8905 | 0.3587 | 1.0120 | 0.3488 | 0.8983 |
| | 0.5 | 0.5079 | 1.4066 | 0.5646 | 1.6820 | 0.4925 | 1.4998 | 0.3643 | 0.9674 | 0.3742 | 0.9044 | 0.3582 | 0.9088 |
| | 0.75 | 0.5327 | 1.3574 | 0.6009 | 1.5105 | 0.5077 | 1.4032 | 0.3942 | 1.2237 | 0.4034 | 1.2992 | 0.3809 | 0.9664 |
| 24 | 0.25 | 0.5393 | 1.1736 | 0.5730 | 1.2878 | 0.5161 | 1.1983 | 0.4190 | 0.9343 | 0.4259 | 0.9161 | 0.4092 | 0.9483 |
| | 0.5 | 0.6034 | 1.2514 | 0.6547 | 1.9296 | 0.5745 | 1.2461 | 0.4624 | 1.0145 | 0.4725 | 0.9711 | 0.4529 | 0.8980 |
| | 0.75 | 0.5916 | 1.1856 | 0.6411 | 1.3456 | 0.5649 | 1.1663 | 0.4622 | 0.9883 | 0.4864 | 0.9311 | 0.4537 | 0.8884 |
| Total mean | | 0.4733 | | 0.5246 | | 0.4551 | | 0.3412 | | 0.3504 | | 0.3358 | |

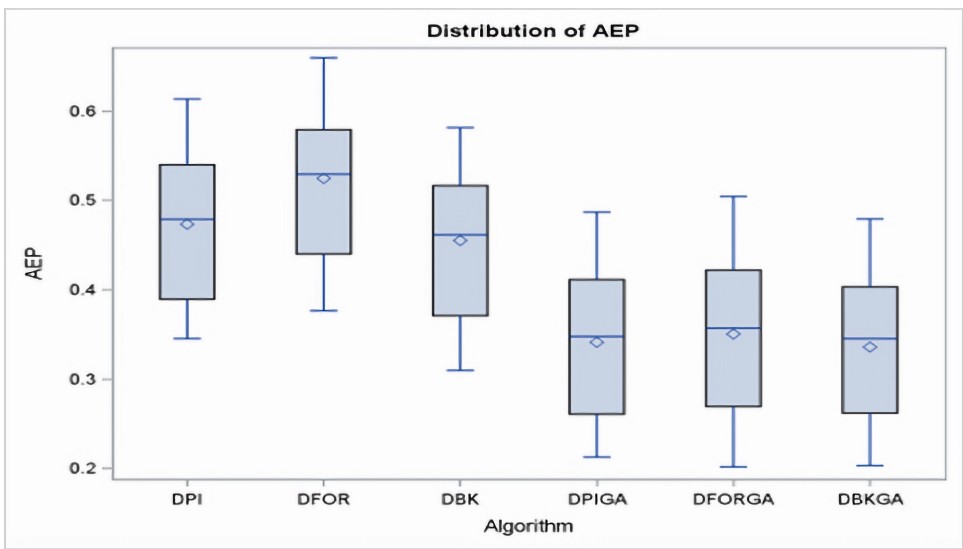

**Figure 2  Boxplots of AEPs for three heuristics and three GAs for small $n$.**

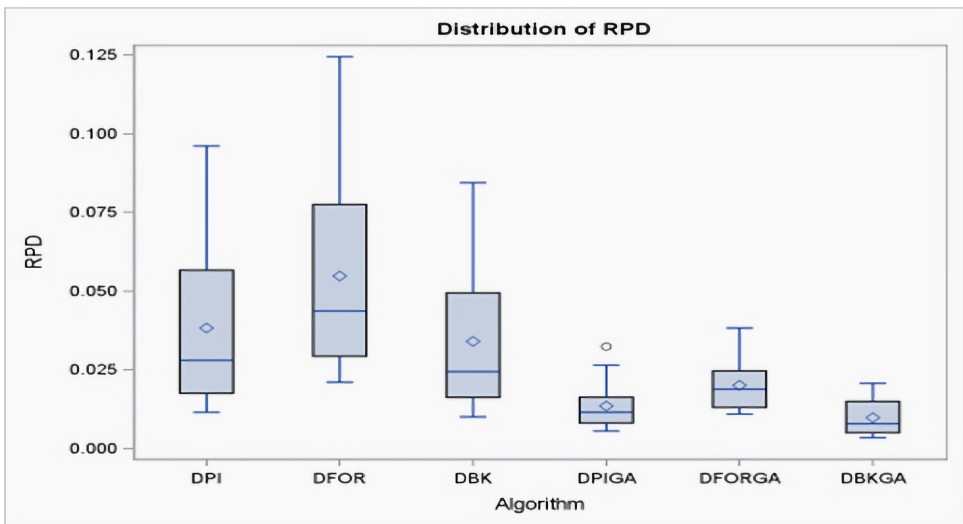

**Figure 3  Boxplots of RPDs for three heuristics and three GAs for large n.**

5 reveal a consistent trend: regardless of the value of $n$, DBKGA consistently achieves the lowest average RPD among the three heuristics (DPI, DFOR, and DBK) and the two GAs (DPIGA, DFORGA). Specifically, the average RPD values are 0.0383, 0.0548, and 0.0341 for DPI, DFOR, and DBK, respectively, and 0.0136, 0.0201, and 0.0099 for DPIGA, DFORGA, and DBKGA, respectively.

Additionally, box plots are presented in Fig. 3 to visually depict the mean RPD for the heuristics and GAs. These plots clearly show that the overall range and interquartile ranges of the mean RPDs are broader for DPI, DFOR, and DBK than for DPIGA, DFORGA, and DBKGA. As indicated in Table 5, as n increases from 60 to 240, the average RPD values exhibit a declining trend.

Regarding computational times, Fig. 4 displays violin plots that visualize the distribution of average CPU time for both the three heuristic algorithms and the three GAs under the conditions of large $n$. These plots clearly indicate that the overall range and interquartile ranges of average CPU times for DPIGA, DFORGA, and DBKGA are broader compared to DPI, DFOR, and DBK. Specifically, the average CPU time values are 0.0196, 0.0195, and 0.0197 for DPI, DFOR, and DBK, respectively, while they are 2.5596, 2.6028, and 2.5983 for DPIGA, DFORGA, and DBKGA, respectively. We do not provide CPU time reports for these six methods in the case of small $n$ because they are less than one second of CPU time.

Below, we delve into the statistical significance of distinctions among the three heuristic approaches and the three GA algorithms. Initially, we applied an analysis of variance (ANOVA) technique utilizing a linear model on the AEP data for small '$n$' or RPD data for large '$n$' within SAS 9.4. The ANOVA, shown in Table 6 and 7, revealed that the model was significant for both small '$n$' and large '$n$'. Additionally, the algorithm and job-size ($n$) factors exhibited significance for both small '$n$' with an F test value of 456.67 and five degrees of freedom and for RPD with an F test value of 50.13 and five degrees of freedom. It is also noticeable that the factors of weight ($\theta$) for the objective function and number

**Table 5** Summary of RPDs of three heuristics (DPI, DFOR, and DBK) and three GAs (DPIGA, DFORGA, and DBKGA) (large n).

| n | m | DPI | | DFOR | | DBK | | DPIGA | | DFORGA | | DBKGA | | Total FS |
|---|---|---|---|---|---|---|---|---|---|---|---|---|---|---|
| | | mean | max | mean | max | mean | max | mean | max | mean | max | mean | max | |
| 60 | 4 | 0.0864 | 0.3481 | 0.1162 | 0.3768 | 0.0786 | 0.3346 | 0.0251 | 0.1814 | 0.0350 | 0.2119 | 0.0177 | 0.1285 | 300 |
| | 6 | 0.0780 | 0.2712 | 0.1001 | 0.3603 | 0.0688 | 0.2885 | 0.0239 | 0.1475 | 0.0301 | 0.1507 | 0.0190 | 0.1046 | 300 |
| 120 | 6 | 0.0368 | 0.1408 | 0.0545 | 0.1774 | 0.0320 | 0.1337 | 0.0136 | 0.1155 | 0.0190 | 0.1179 | 0.0112 | 0.0638 | 300 |
| | 10 | 0.0348 | 0.1163 | 0.0509 | 0.1575 | 0.0303 | 0.1061 | 0.0133 | 0.0778 | 0.0205 | 0.1062 | 0.0101 | 0.0735 | 300 |
| 180 | 6 | 0.0221 | 0.0843 | 0.0352 | 0.1276 | 0.0198 | 0.0845 | 0.0104 | 0.0685 | 0.0165 | 0.0737 | 0.0063 | 0.0456 | 300 |
| | 10 | 0.0204 | 0.0585 | 0.0335 | 0.1076 | 0.0185 | 0.0588 | 0.0084 | 0.0530 | 0.0163 | 0.0782 | 0.0063 | 0.0387 | 300 |
| 240 | 6 | 0.0154 | 0.0555 | 0.0256 | 0.0901 | 0.0136 | 0.0597 | 0.0076 | 0.0551 | 0.0124 | 0.0557 | 0.0043 | 0.0330 | 300 |
| | 10 | 0.0124 | 0.0471 | 0.0224 | 0.0852 | 0.0109 | 0.0479 | 0.0060 | 0.0319 | 0.0112 | 0.0553 | 0.0042 | 0.0262 | 300 |
| | $\theta$ | | | | | | | | | | | | | |
| 60 | 0.25 | 0.0771 | 0.2836 | 0.1028 | 0.3270 | 0.0696 | 0.2745 | 0.0212 | 0.1082 | 0.0303 | 0.1614 | 0.0192 | 0.1380 | 200 |
| | 0.50 | 0.0826 | 0.3361 | 0.1082 | 0.4000 | 0.0763 | 0.3409 | 0.0259 | 0.1940 | 0.0317 | 0.2023 | 0.0195 | 0.1062 | 200 |
| | 0.75 | 0.0868 | 0.3093 | 0.1134 | 0.3786 | 0.0753 | 0.3193 | 0.0264 | 0.1913 | 0.0357 | 0.1801 | 0.0163 | 0.1055 | 200 |
| 120 | 0.25 | 0.0357 | 0.1433 | 0.0537 | 0.1625 | 0.0303 | 0.1325 | 0.0127 | 0.0902 | 0.0184 | 0.1274 | 0.0094 | 0.0696 | 200 |
| | 0.50 | 0.0339 | 0.1096 | 0.0491 | 0.1504 | 0.0303 | 0.1053 | 0.0136 | 0.0918 | 0.0198 | 0.0964 | 0.0105 | 0.0526 | 200 |
| | 0.75 | 0.0379 | 0.1328 | 0.0554 | 0.1895 | 0.0328 | 0.1220 | 0.0141 | 0.1080 | 0.0211 | 0.1123 | 0.0121 | 0.0838 | 200 |
| 180 | 0.25 | 0.0206 | 0.0717 | 0.0337 | 0.1128 | 0.0188 | 0.0661 | 0.0084 | 0.0586 | 0.0140 | 0.0616 | 0.0062 | 0.0369 | 200 |
| | 0.50 | 0.0230 | 0.0674 | 0.0368 | 0.1200 | 0.0206 | 0.0765 | 0.0100 | 0.0640 | 0.0191 | 0.0841 | 0.0060 | 0.0429 | 200 |
| | 0.75 | 0.0202 | 0.0752 | 0.0325 | 0.1199 | 0.0181 | 0.0723 | 0.0098 | 0.0596 | 0.0161 | 0.0821 | 0.0066 | 0.0466 | 200 |
| 240 | 0.25 | 0.0140 | 0.0461 | 0.0231 | 0.0804 | 0.0116 | 0.0517 | 0.0075 | 0.0383 | 0.0114 | 0.0497 | 0.0045 | 0.0286 | 200 |
| | 0.50 | 0.0146 | 0.0568 | 0.0255 | 0.0804 | 0.0136 | 0.0610 | 0.0067 | 0.0465 | 0.0126 | 0.0589 | 0.0042 | 0.0289 | 200 |
| | 0.75 | 0.0131 | 0.0509 | 0.0234 | 0.1022 | 0.0115 | 0.0486 | 0.0063 | 0.0457 | 0.0114 | 0.0580 | 0.0040 | 0.0315 | 200 |
| Total mean | | 0.0383 | | 0.0548 | | 0.0341 | | 0.0136 | | 0.0201 | | 0.0099 | | |

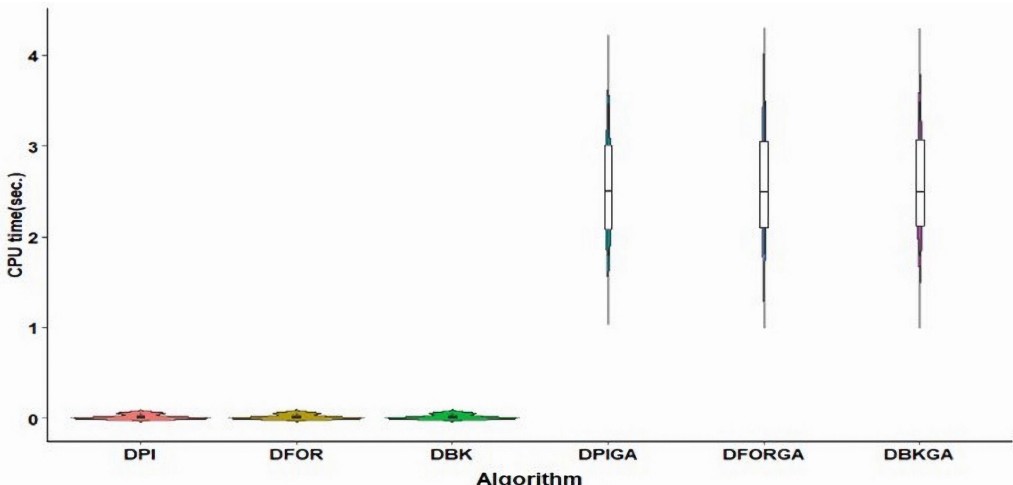

**Figure 4** Violin plots of CPU time for three heuristics and three GA algorithms (large n).

**Table 6 ANOVA for small n.**

| Source | DF | Type I SS | Mean Squared | F value | Pr>F |
|---|---|---|---|---|---|
| Algorithm | 5 | 0.78956555 | 0.15791311 | 456.67 | <.0001 |
| n | 3 | 0.91580969 | 0.30526990 | 882.82 | <.0001 |
| $\theta$ | 2 | 0.04142559 | 0.02071280 | 59.90 | <.0001 |
| m | 4 | 0.00987289 | 0.00246822 | 7.14 | <.0001 |

**Table 7 ANOVA for big n.**

| Source | DF | Type I SS | Mean Squared | F value | Pr>F |
|---|---|---|---|---|---|
| Algorithm | 5 | 0.03502865 | 0.00700573 | 50.13 | <.0001 |
| n | 3 | 0.04215221 | 0.01405074 | 100.55 | <.0001 |
| $\theta$ | 2 | 0.00010492 | 0.00005246 | 0.38 | 0.6878 |
| m | 4 | 0.00044112 | 0.00011028 | 0.79 | 0.5342 |

of orders ($m$) are significant in Table 6 for small '$n$', whereas they did not demonstrate significance for the large '$n$' case shown in Table 7. The rationale behind this observation is that the effect of '$n$' outweighs the values of weight ($\theta$) for the objective function or the number of orders ($m$) in the context of the large '$n$' case.

The normality assumption of the linear model was validated for both the AEP and RPD data. This was evidenced by the Kolmogorov–Smirnov normality test, which produced $p$ values exceeding 0.05 (with statistic values of 0.053336 for small '$n$' and 0.058442 for large '$n$'). Additionally, the AEP and RPD Tukey grouping for means of algorithms were also tested. Figure 5 shows that the algorithms can be ranked into five groups. The five groups range from the worst to the best as DFOR, DPI, DBK, DFORGA, and (DPIGA, DBKGA) for the AEPs, shown in Fig. 5 (left) as well as the four groups from the worst to the best as

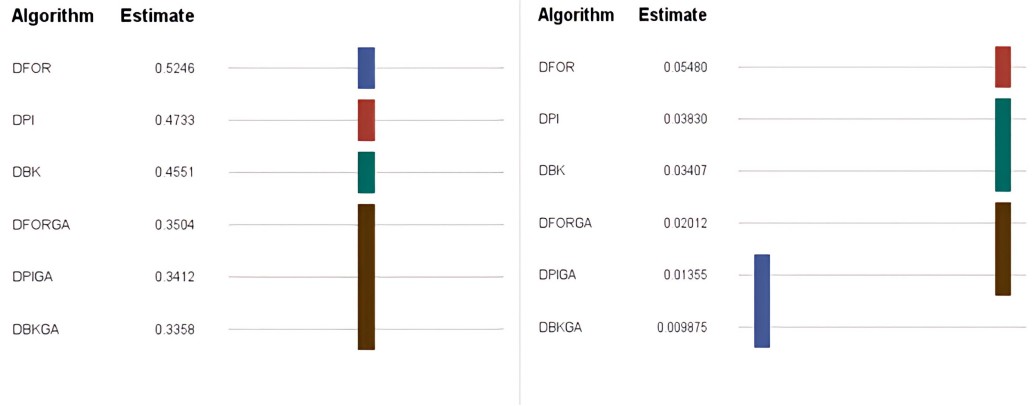

**Figure 5** **The AEP and RPD Tukey grouping for means of algorithms (Alpha = 0.05).** Means covered by the same bar are not significantly different.

DFOR, (DPI, DBK), (DFORGA, DPIGA), and (DPIGA, DBKGA) for the RPDs shown in Fig. 5 (right).

The three heuristics (DPI, DFOR, and DBK) rely on predefined rules or strategies to find solutions quickly. While effective in some cases, they may get stuck in local optima and fail to explore the entire solution space. On the other hand, the three GAs (DPIGA, DFORGA, and DBKGA) employ population-based searches and a genetic-inspired selection, crossover, and mutation process, which allows them to explore a broader range of the solution space, potentially discovering better solutions. The advantages of genetic algorithms over heuristics in setup time optimization often stem from their ability to handle complex relationships, explore diverse solution spaces, and converge toward globally optimal solutions by using evolutionary mechanisms. Overall, it can be confirmed that the DBKGA is the best approach with the smallest value of AEP or RPD among all six proposed algorithms.

## CONCLUSIONS AND SUGGESTIONS

In practical settings, a machine incurs a setup time whenever it switches from processing one product to another. Scheduling jobs while taking into account setup times and associated costs has attracted significant attention in both the manufacturing and service sectors, driving extensive research endeavors. While prior studies on customer order scheduling often focused on jobs to be processed across multiple machines, they frequently overlooked the critical factor of setup time. This study addresses a sequence-dependent bi-criterion scheduling problem, integrating considerations for order delivery. The primary aim is to pinpoint an optimal solution that minimizes the combined makespan and weighted sum of completion times for all specified orders.

To tackle this formidable challenge, we propose a branch-and-bound method incorporating two dominance properties and a lower bound. This approach is finely tuned to yield optimal solutions in scenarios involving a limited number of jobs, exemplified by cases with $n = 12$, 16, 20, and 24. Demonstrating notable efficacy, this method excels for

up to $n = 24$ jobs, accommodating diverse combinations of order quantities and objective weights. For both small $n$ (=12, 16, 20, and 24) and large $n$ (= 60, 120, 180, and 240) tested instances, we introduce a tailored heuristic approach. This heuristic is complemented by three strategic techniques: pairwise interchange, extraction and forward-shifted reinsertion, and extraction and backward-shifted reinsertion (DPI, DFOR, and DBK). Additionally, we employ three heuristic-based genetic algorithms (DPIGA, DFORGA, and DBKGA) to obtain reliable approximate solutions.

Notably, it is evident that DBKGA performed significantly better than the five other proposed heuristics for both small and large test cases. A potential direction for future research could involve expanding the model to a sequence-dependent flow shop setting that incorporates order deliveries. Furthermore, it may be worthwhile to investigate extending this framework to minimize the total tardiness of all fulfilled orders. As another future topic, we can also consider that the machine can break down or extend this model to the multiple-machine setting.

## ACKNOWLEDGEMENTS

We thank the Academic Editor and two referees for their positive and useful comments.

### Funding

This research received support from the National Natural Science Foundation of China, grant number 72271048, and was also funded by the National Science and Technology Council of Taiwan, under Grant No. NSTC 112-2221-E-035-060-MY2. The funders had no role in study design, data collection and analysis, decision to publish, or preparation of the manuscript.

### Grant Disclosures

The following grant information was disclosed by the authors:
The National Natural Science Foundation of China: 72271048.
The National Science and Technology Council of Taiwan: NSTC 112-2221-E-035-060-MY2.

### Competing Interests

The authors declare there are no competing interests.

### Author Contributions

- Jian-You Xu conceived and designed the experiments, analyzed the data, authored or reviewed drafts of the article, pay the charge once paper is accepted, and approved the final draft.
- Win-Chin Lin conceived and designed the experiments, analyzed the data, authored or reviewed drafts of the article, and approved the final draft.

- Kai-Xiang Hu performed the experiments, performed the computation work, prepared figures and/or tables, authored or reviewed drafts of the article, and approved the final draft.
- Yu-Wei Chang performed the experiments, performed the computation work, prepared figures and/or tables, authored or reviewed drafts of the article, and approved the final draft.
- Wen-Hsiang Wu performed the computation work, prepared figures and/or tables, authored or reviewed drafts of the article, and approved the final draft.
- Peng-Hsiang Hsu performed the computation work, prepared figures and/or tables, authored or reviewed drafts of the article, and approved the final draft.
- Tsung-Hsien Wu conceived and designed the experiments, prepared figures and/or tables, authored or reviewed drafts of the article, and approved the final draft.
- Chin-Chia Wu analyzed the data, authored or reviewed drafts of the article, and approved the final draft.

## Data Availability

The code is available in the Supplemental File.

## Supplemental Information

Supplemental information for this article can be found online at http://dx.doi.org/10.7717/peerj-cs.1763#supplemental-information.

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
