# Peer review of "A bi-criterion sequence-dependent scheduling problem with order deliveries"

_PeerJ Computer Science, doi:10.7717/peerj-cs.1763_

## Round 0.1 · original submission · Major Revisions

Dear authors,

Your paper has been reviewed by two reviewers who asked for revisions of the paper. Please revise the paper according to comments by reviewers, mark all changes in the new version of the paper, and provide cover letter with replies to them point to point.

**Language Note:** PeerJ staff have identified that the English language needs to be improved. When you prepare your next revision, please either (i) have a colleague who is proficient in English and familiar with the subject matter review your manuscript, or (ii) contact a professional editing service to review your manuscript. PeerJ can provide language editing services - you can contact us at [email protected] for pricing (be sure to provide your manuscript number and title). – PeerJ Staff

·

Basic reporting

1. The introduction could provide more background on related scheduling problems and applications to motivate the study. Elaborate on why optimizing setup times is an important issue.

2. Explain all acronyms when first introduced, such as COSP, SDST, etc.

3. Provide more details on the algorithms in the methods section, such as pseudocode for the heuristics.

Experimental design

4. Include a flow chart or diagram to visually summarize the overall solution methodology.

5. Elaborate on the parameter tuning process for the genetic algorithms. How were the parameter values narrowed down?

6. Justify why certain parameter values were chosen over others through statistical tests or plots.

Validity of the findings

7. Provide more analysis between the results of the heuristics versus the genetic algorithms. What specifically led GA to outperform?

8. Conduct statistical tests like ANOVA to verify the significance of differences between algorithms.

9. Discuss limitations of the model assumptions, data, or methodology.

Additional comments

10. Suggest future work such as testing other metaheuristics, additional problem criteria, or real-world applications.

Reviewer 2 ·

Basic reporting

The paper addresses a sequence-dependent bi-criterion scheduling problem, incorporating order delivery considerations. This is a very important research topic fully in the scope of the journal. The proposed paper is written in English language using clear, unambiguous, technically correct text. The structure of the paper is clear and appropriate for the journal. The article includes sufficient introduction and background to demonstrate how the work fits into the broader field of knowledge. However, I think that this study's hypothesis, research question, and study gaps should be explained in more detail. The authors should make the research limitations and prospect of this work clearer.

Experimental design

The authors use different methodological approaches to solve defined scheduling problem, including exact methods such as B&B algorithm, three heuristic methods (DPI, DFOR, and DBK), and three heuristic-based GAs (DPIGA, DFORGA, and DBKGA). It is well-known that B&B algorithm can ensure the optimality and efficiency for a limited number of jobs. Also, GAs yield optimal solutions for both small and large test cases, but for optimal input parameters (for example size of population, mutation parameters, crossover parameter…). These are not the conclusions of this paper only, but also of numerous previous papers in this field. In that context it is not clear what is the main contribution of this paper?

Validity of the findings

In this paper computational simulations are done using randomly generated data (24 cases for both small and large job numbers). A potential direction for future research could involve verification of conclusions using real study data.

Additional comments

No additional comments.

Cite this review as

---

## Round 0.2 · accepted · Accept

Dear authors,

Your revised version of the paper has been reviewed by a reviewer who previously asked for major revisions. He recommended accepting your paper. I also evaluated your revisions and I concur.

·

Basic reporting

The paper is improved well and can be accepted

Experimental design

-

Validity of the findings

-

Additional comments

-